Acaricidal and insecticidal activities of entomopathogenic nematodes combined with rosemary essential oil and bacterium-synthesized silver nanoparticles against camel tick, Hyalomma dromedarii and wax moth, Galleria mellonella

http://orcid.org/0000-0002-2592-8281 Albogami Bander 1
Darwish Hadeer 2
Alghamdi Akram 1
http://orcid.org/0000-0003-2231-3983 Darwish Ahmed BahaaEldin 3
Al-Otaibi Wafa Mohammed 1
A. Osman Mohamed 4 5
M. Al Dhafar Zamzam 4 5
Alkhaibari Abeer Mousa 6
Mashlawi Abadi M. 7
Baakdah Fadi 8 9
Noureldeen Ahmed 1 a.noureldeen@tu.edu.sa
1 Department of Biology, College of Science, Taif University , Taif , Saudi Arabia
2 Department of Biotechnology, College of Sciences, Taif University , Taif , Saudi Arabia
3 Department of Zoology, Suez University , Suez , Egypt
4 Department of Biology, College of Science, Imam Abdulrahman Bin Faisal University , Dammam , Saudi Arabia
5 Basic and Applied Scientific Research Center (BASRC), Imam Abdulrahman Bin Faisal University , Dammam , Saudi Arabia
6 Department of Biology, Faculty of Science, University of Tabuk , Tabuk , Saudi Arabia
7 Department of Biology, College of Science, Jazan University , Jazan , Saudi Arabia
8 Department of Medical Laboratory Sciences, Faculty of Applied Medical Sciences, King Abdulaziz University , Jeddah , Saudi Arabia
9 Special Infectious Agents Unit, King Fahd Medical Research Center, King Abdul Aziz University , Jeddah , Saudi Arabia
Tomar Mahendra
Electronic publication date: 2025 Jan 14
Publication date: 2025
Volume: 13
Electronic Location ID: e18782
Received 2024 Sep 2; Accepted 2024 Dec 9
Copyright: © 2025 Albogami et al.
Copyright year: 2025
Copyright holder: Albogami et al.
License: This is an open access article distributed under the terms of the Creative Commons Attribution License, which permits unrestricted use, distribution, reproduction and adaptation in any medium and for any purpose provided that it is properly attributed. For attribution, the original author(s), title, publication source (PeerJ) and either DOI or URL of the article must be cited.
License URL: https://creativecommons.org/licenses/by/4.0/

Keywords: Entomopathogenic nematodes, Essential oil, Nanoparticles, Hyalomma dromedarii, Galleria mellonella, Bioactivity

Funding: Taif University, Saudi Arabia TU-DSPP-2024-155 This research was funded by Taif University, Saudi Arabia, Project No. (TU-DSPP-2024-155). The funders had no role in study design, data collection and analysis, decision to publish, or preparation of the manuscript.

==============================
An innovative approach to ticks and insect pests management is necessary to mitigate the challenges posed by the indiscriminate use of chemical pesticides, which can lead to resistance development and environmental pollution. Despite their great potential, biological control agents have significant manufacturing, application, and stability limitations. Currently, using phytochemicals, biosynthesized nanoparticles, and bioagents to get rid of arthropods might be a good alternative that would make farmers less worried about residues and resistance. The present investigation was carried out to determine for the first time the in vitro acaricidal and insecticidal efficacies of endogenous two entomopathogenic nematodes (EPNs), Heterorhabditis indica and Steinernema sp. combined with either Proteus mirabilis-synthesized silver nanoparticles or Rosmarinus officinalis essential oil against the camel tick, Hyalomma dromedarii larvae and females, and greater wax moth, Galleria mellonella larvae as well. We also determined the potential effects of these treatments on the biological characteristics of H. dromedarii’s engorged females. We further investigated R. officinalis essential oil (EO) profiling and nanoparticle (AgNPs) characterization. All the evaluated combinations demonstrated synergistic effects on the larvae of G. mellonella and H. dromedarii, as well as on engorged females. When H. indica was mixed with EO or AgNPs, it worked well than when Steinernema sp. was mixed with EO or AgNPs. This was shown by the highest number of tick and insect mortalities and the lowest lethal concentration (LC50) values. One day after G. mellonella was exposed to H. indica (1,000 infective juveniles (IJs)) together with EO at 60 or 40 mg/mL, all tested individuals died. We obtained the same results when H. dromedarii females exposed to the same level of EPN with 60 mg/mL EO, and when H. dromedarii larvae treated with H. indica at 500 IJs + EO at 25 mg/mL. Treatments altered all biological parameters of engorged females, revealing extremely noticeable differences between the treated and untreated groups. Gas chromatography–mass spectrometry (GC-MS) analysis identified a total of 28 compounds in the R. officinalis EO. Visual observation showed a color change from yellow to dark brown for AgNPs biosynthesized from P. mirabilis; the transmission electron microscopy (TEM) image and ultraviolet–visible (UV-Vis) spectrum showed well-dispersed particles with a diameter of 5–45 nm; and the greatest surface plasmon peaked at 320 nm. The results demonstrated the high efficacy of combining EPN, H. indica, with EO to control tick and insect pests. This is due to its acaricidal activity on different stages of H. dromedarii, including larvae and engorged females, and its larvicidal effect on G. mellonella.

Introduction

Ticks are ectoparasites that feed on vertebrates, resulting in significant blood loss and decreased weight gain in infested livestock (Mukhebi et al., 1999), and they are crucial in the spread of numerous fatal diseases to both livestock and humans (Raoult et al., 2001; Parola, 2004). The camel industry faces a significant challenge from ixodid ticks due to their ability to transmit diseases, as well as their tendency to inflict economic losses in infected animals (Jabbar et al., 2015). However, research revealed that over 20 species of ixodid infest camels (Banaja & Ghandour, 1994), making their active control crucial. The most common species among them are ticks belonging to the genus Hyalomma, which could serve as vectors for Theileria spp., Babesia spp., and Anaplasma spp. (Fard et al., 2012; Barghash & Hafez, 2016; Alanazi et al., 2018).

Hyalomma dromedarii (Acari: Ixodidae) was predominantly tick-infested camels. Mostly, ticks and mites infest camels, but they also affect cattle (Barker & Murrell, 2004). About 95.6% of Sinai camels, 89% of Sudan camels, and 57.13% of Benha and Belbis (Egypt) camels were infested (Ashjaran & Sheybani, 2019; Al-Ghamdi, Aly & Sheshtawi, 2020). Saudi Arabia is one of the nations where the camel population has grown rapidly in recent years. As of 2017, the country had about 500,000 camels, with the largest percentage living in Riyadh Province (FAO, 2019). Saudi Arabia has documented numerous species, including different Hyalomma spp., infesting camels and cattle (Al-Afaleq et al., 2018).

Currently, we use various synthetic acaricides like avermectin, deltamethrin, and cypermethrin (Al Thabiani et al., 2021) to manage H. dromedarii. However, acaricide-resistant tick strains developed, polluted the environment, contaminated meat and dairy products, and the high cost as a result of the ongoing usage of these acaricides made them inaccessible by small-scale farmers (Gupta & Gupta, 2018; Singh, Abhijit & Harkirat, 2019). The existence of several tick species that present diverse antigens also limits vaccination efficiency (de la Fuente & Estrada-Pena, 2019). Consequently, there was a worldwide movement to assess and propose environmentally friendly and efficient agents as a substitute since they were accessible, affordable, and had low levels of toxicity, resistance, and environmental pollution.

Biological acaricides are safe, cost-effective, and biodegradable (Karthi et al., 2020). An integrated tick management program (ITMP) requires a variety of techniques with different modes of action for successful execution. This will optimize effectiveness while lowering the possibility that target pests will develop resistance to the chosen treatments, which may include chemical acaricides (de Leon, 2017; Rodriguez-Vivas, Jonsson & Bhushan, 2018).

The only species in the genus Galleria is the greater wax moth (Hamza & Sayed, 2019), often known as the honeycomb moth (Galleria mellonella L.; Lepidoptera: Pyralidae). The caterpillar larvae feed on the honeycomb found inside bee hives, so they commonly known as wax worms. However, certain traits that G. mellonella possesses make it a suitable choice for researching host-pathogen relationships. It is an excellent, inexpensive, simple, and quick in vivo model to evaluate the pathogenicity of microorganisms, the toxicity of different compounds, and also to study nanoparticle interactions with living systems, and has numerous advantages over other animal models as well (Cutuli et al., 2019). We can run their larvae at the human core body temperature of 37 °C, and they display a response to the innate immune system akin to that of mammals, leading to pathogen elimination through comparable mechanisms (Wojda et al., 2020; Noureldeen et al., 2019; Andrea, Krogfelt & Jenssen, 2019).

Entomopathogenic nematodes (EPNs) of the families Steinernematidae and Heterorhabditidae offer effective substitutes to the acaricides and insecticides currently in use because many investigators have found encouraging results, with some isolates exhibiting acaricidal and insecticidal activities on certain tick and insect species or developmental stages (Monteiro & Prata, 2015; EL Roby, Rezk & Shamseldean, 2018; Alotaibi et al., 2021; Noureldeen et al., 2022). Together with the help of mutualistic bacteria, EPNs kill ticks and other insects. Photorhabdus and Xenorhabdus bacteria and Heterorhabditis and Steinernema nematodes live together in harmony (Gaugler & Kaya, 1990). The only free-living stage, infective juvenile (IJ), enters hosts by spiracles, the mouth, anus, and occasionally the cuticle. They kill the host within 48 h after releasing their bacterial symbionts into the hemocoel (Koppenhofer, 2007).

According to previous research (El-Sadawy, Zayed & El-Shazly, 2008), heterorhabditid strains have a greater influence on H. dromedarii engorged females than do stienernematid strains, with mortality rates ranging from 12–92% for the former and 4-88% for the latter. Furthermore, it was mentioned that G. mellonella, which is utilized as a model to investigate EPNs, is extremely vulnerable to infection by IJs (Ehlers & Shapiro-Ilan, 2005). Research has indicated that specific EPN isolates exhibit a stronger attraction to G. mellonella when compared to other insect species (Koppenhöfer & Fuzy, 2008). According to Bisch et al. (2015), EPNs form a symbiotic association with Enterobacteriaceae bacteria, which may act as an effective biocontrol alternative for G. mellonella.

Nanotechnology, a rapidly developing field, creates biomaterials with precise size and shape. Nanoparticles (NPs) are widely used in numerous disciplines and incorporated into a wide range of commercial products, as well as biomedical and veterinary applications, due to their small size (de Oliveira et al., 2021; Abdel-Ghany et al., 2022). At numerous stages of the parasite’s life cycle, metallic and non-metallic nanoparticles have demonstrated significant potential for quickly causing toxicity and lowering lethal doses (Benelli et al., 2017; Esmaeilnejad et al., 2018). Currently, nanobiotechnology is thought to be a revolutionary approach to tick control. Silver is a valuable and widely available metal, and studies have suggested that its nanoparticles may have anti-tick and anti-insect properties because they are preferable to other nanosized metal particles (Norouzi et al., 2019).

Today, a novel technique for producing nanomaterials is facilitating the development of biotic systems, particularly microorganisms (Saha et al., 2017). Because biological methods are more economical, sustainable, and ecologically benign than chemical and physical processes, they are preferable for synthesizing nanoparticles. As a result, nanotechnology scientists are interested in using prokaryotic organisms and their exudate to produce NPs (Sengani, Grumezescu & Rajeswari, 2017). Numerous microorganisms, including yeasts, actinomycetes, fungi, and bacteria, can create NPs by extracellular or intracellular pathways. Several bacteria, including Rhodopseudomonas capsulate, Bacillus subtilis, Bacillus megaterium, Pseudomonas aeruginosa, Brevibacterium casei, Escherichia coli DH5a, and Stenotrophomonas maltophilia, were used to produce silver nanoparticles (Rajeshkumar et al., 2016). Previous studies have documented the negative effects of the active components of the gram-negative bacteria Proteus mirabilis on the tick Hyalomma marginatum (Hendry & Rechav, 1981). Recently, Proteus mirabilis is considered one of the most important bacteria that have been utilized for synthesizing silver nanoparticles (AgNPs), showing their antioxidant ability in vitro, and being as antimicrobial and disinfectant agents (Elhariry et al., 2018; Yasr & AL-Ramahy, 2022).

As an alternative to pesticides (Abdel-Meguid et al., 2022), botanicals like essential oils (EOs) show promise because they are safe for non-target organisms, break down quickly, have a complex makeup that keeps resistance from developing, and have effects that kill adults, eggs, larvae, and other pests (Baz et al., 2022). Plant products have benefits, but they also have disadvantages, such as inconsistent effectiveness, chemical instability, and little residual activity, often caused by photosensitivity or excessive volatility (George et al., 2014). The literature reveals that researchers have globally examined 356 plant species from 105 families against 31 tick species (Nwanade et al., 2020). In previous studies, essential oils of different plants revealed acaricidal activity against different tick species, such as Lavandula angustifolia, Eucalyptus globulus, Mentha piperita, Artemisia annua, Rosmarinus officinalis, and Thymus vulgaris oils against Rhipicephalus microplus and Hyalomma spp. (Djebir et al., 2019; Valcarcel et al., 2021).

Worldwide, there are no studies that have examined the interaction effects of EPNs with either nanoparticles or plant products against H. dromedarii. Therefore, this study was conducted for the first time to assess the in vitro acaricidal and insecticidal efficacies of locally two EPNs, Heterorhabditis indica and Steinernema sp. integrated with either Proteus mirabilis-synthesized silver nanoparticles or Rosmarinus officinalis essential oil, against the camel tick Hyalomma dromedarii and greater wax moth Galleria mellonella, as well as determine their potentials on biological parameters of H. dromedarii-engorged females.

Materials and Methods

Ticks

Hyalomma dromedarii engorged females (about 20 not parasitized individuals, homogenized in size, age, and shape per camel) were hand-picked from naturally infested camels without having received any acaricide treatment on a farm situated in Taif province, Saudi Arabia. The identification of ticks was made under stereomicroscope according to Estrada-Peña (2015). After being collected in small plastic boxes with ventilation holes, ticks were transferred to the Parasitology Laboratory, Department of Biology, College of Science, Taif University. Until oviposition, engorged females were kept at 25 ± 1 °C and 75–80% RH in a plastic cup after being properly washed in water and dried. To obtain larvae, eggs were collected daily for 20 days, mixed thoroughly, and then put into glass tubes (12 × 75 mm) and maintained under situations of similar temperature and relative humidity.

Insects

Galleria mellonella larvae were obtained from wax combs of naturally infested honey bee colonies from the private apiary in the Taif governorate, Taif, Saudi Arabia. The continuation of G. mellonella culture was accomplished according to Admella & Torrents (2022) with little modifications, by placing around 60 larvae into containers containing naturally darkened honeycomb medium in non-aseptic conditions and fed until the pupal stage. The culture was raised in dark conditions at 28 ± 2 °C with RH 60 ± 5%. Once the adult moths emerged, the males and females were transferred together into other plastic containers that were covered with white tissue paper and allowed to breed and lay eggs for a few days. The eggs were collected and transferred to artificial diets (25% honey, 15% wheat flour, 15% corn flour, 11% powdered milk, 15% infant cereal, 6% brewer’s yeast, and 13% glycerol) used as food by the newly emerged larvae till pupation, then the steps were routinely repeated for 2–3 generations. The G. mellonella was morphologically identified in accordance with Kwadha et al. (2017).

Entomopathogenic nematodes

Two native isolates of EPNs, i.e., Steinernema sp. TUAN1 (OQ309179), and Heterorhabditis indica NEM-24 (OP578200) were used in this study. The nematodes were maintained on the last instar larvae of G. mellonella based on the procedure of Kaya & Stock (1997). Every day, the emerging IJs of EPNs were removed from nematode traps, which were then kept in 40 mL cell culture bottles (20 mL aliquots) at 10 °C. IJs were all stored in a refrigerator at 8 °C and used for up to 2 weeks after they were emerged in the bioassays. In order to prepare the suspensions, ten aliquots (10 μL) of each stored IJs were counted. The average number of IJs per sample was then determined, and the suspensions were adapted to the required concentrations based on this average.

Rosmarinus officinalis essential oil

Plant materials

Rosemary, R. officinalis (Lamiaceae) plants 30 cm long and 2 months old were obtained from cultured plants in the greenhouse of College of Science, Taif University, Saudi Arabia. Fresh leaves of R. officinalis were collected, completely rinsed with distilled water, and then kept at room temperature for 2 weeks to air dry.

Essential oil extraction

Dried leaves were ground into a powder using mixer blinder (MRC LB20ES). Utilizing a Clevenger-style apparatus, 500 mL of distilled water and 100 g of powdered leaves were hydrodistilated for 3 h. The distillation process was repeated in order to obtain the proper quantity of oil for subsequent usage. Following filtering and drying on anhydrous sodium sulfate, the essential oil distillates were stored at −4 °C until analysis (Hussain et al., 2008). For bioassays, an aqueous solution having EO of R. officinalis at 1–6% (10–60 mg/mL) with 2% Tween 20 (20 mg/mL) was prepared. Next, the mixture was vortexed until it became homogeneous.

GC–MS analysis of EO

A 6890N Network gas chromatography–mass spectrometry (GC-MS) system from Agilent Technologies (Little Falls, CA, USA) was used to analyze the essential oil (Hussain et al., 2008). It was supplied with an Agilent Technologies 5975 inert XL Mass selective detector and an Agilent Technologies 7683B series auto injector. Molecules were separated on HP-5 MS capillary columns (30 m × 0.25 mm i.d., 0.25 µm film thickness; Little Falls, CA, USA). Working conditions included a split ratio of 100:1, 1.0 µL of sample injected in the split mode, ionization energy of 70 eV for GC-MS detection, 1 mL/min helium carrier flow rate, temperatures of 220 °C and 290 °C for the MS transfer line and injector, respectively, and mass scanning between 50 and 550 m/z. By comparing their retention indices to those of MS data (Agilent Technologies, 7th edition) and the NIST mass spectrum library, the oil constituents were identified.

Proteus mirabilis bacterium-synthesized silver nanoparticles

Bio-synthesis of AgNPs

Silver nanoparticles (AgNPs) were biosynthesized using the procedure of Fayez, El-Deeb & Mostafa (2017). P. mirabilis bacterium was cultured in a 500 mL Erlenmeyer flask with nutrient broth that contained 5 g of peptone, 3 g of yeast extract, and 5 g of NaCl per liter. The flasks were incubated in a shaker set to 28 °C and 120 rpm for a whole day. Following the incubation period, the culture was centrifuged at 10,000×g, and approximately 0.5 g of the wet biomass of P. mirabilis was agitated with 100 mL sterile double-distilled water at 200 rpm for 48 h at 28 °C in a 250 mL Erlenmeyer flask. Then, centrifugation was implemented to get the cell-free supernatant. Silver nanoparticles were synthesized using this filtrate. A final concentration of 1 mM silver ions was obtained for AgNPs production by mixing the silver nitrate (AgNO3) with 100 mL of cell-free filtrate in a 250 mL Erlenmeyer flask. For 24 h, the mixture was incubated in the dark at 37 °C. AgNPs were examined visually, noting a shift in color from pale yellow to brown, and measuring the UV-visible spectrum. In order to verify the complete conversion of Ag+ to AgNPs and ensure purity, the AgNPs solution underwent a 30-min centrifugation at 15,000×g. Using a spectrophotometer and NaCl solution, the clear supernatant was evaluated, and there was no Ag+ retained in the supernatant, according to both tests. At room temperature, three months after AgNPs were biosynthesised, similar spectra of AgNPs verified their stability. Tests on nutrient agar plates confirmed that the AgNPs solution was free of microbial contamination. To achieve the concentrations of 0.5–4 mg/L for further experiments, the AgNPs were diluted with distilled water.

Characterization of AgNPs

Using a Perkin Elmer Lambda 25 spectrophotometer, ultraviolet–visible (UV–Vis) spectroscopy was carried out as described by Al-Harbi et al. (2014). The size, shape, and structure of the nanoparticles were determined using transmission electron microscopy (TEM) equipped with selected area electron diffraction (SAED) and running at 220 kV accelerating voltage (Jeol-2100; Jeol, Tokyo, Japan). On carbon-coated TEM grids, droplets of the silver nanoparticle solution were applied to create TEM samples. The program ImageJ (National Institutes of Health, USA) was used to process the acquired images.

Bioassays

Effect of EO and AgNPs on survival of EPNs

Survival of IJs of the two nematode species under different concentrations of R. officinalis EO or bio-synthesized AgNPs to estimate their compatibility was evaluated in 24-well plates. In each well, 100 IJs in 1 ml of EO with 2% Tween 20 solution (at a concentration of 10, 20, 40, and 60 mg/mL) or AgNPs with distilled water solution (at a concentration of 0.5, 1, 2, and 4 mg/L) were loaded. In the control treatments, 100 IJs were released into a final volume of 1 mL Tween 20 solution at 2% or distiled water in each well. The trials were performed three times, with ten repetitions for each treatment. To avoid moisture loss, the dishes were coated with parafilm and subsequently incubated at 25 °C for 4 days. The mortality of IJs in each well was recorded after 24, 48, 72, and 96 h post-application. Treatments that produced a mortality rate of less than 25% were classified to be harmless or least toxic (de Morais et al., 2016).

Tick larval immersion test

The acaricidal activity of EPNs, R. officinalis EO, and bio-synthesized AgNPs was assessed using a modified larval immersion test (LIT) against 14-days-old H. dromedarii larvae both individually and in combination. The larvae were divided into seven groups with 800 ticks each (200 larvae/concentration), with previously uniform weights (P > 0.05). For the groups that received only EPNs, each group was weighed and then put into 9 × 1.5 cm petri dishes that had previously been lined with two sheets of filter paper, then 1.0 mL of the suspension comprising 150, 300, 600, and 1,200 IJs was poured to each dish. In the groups treated with R. officinalis EO or AgNPs, 800 larvae (for each group) were immersed for 5 min in a solution (10 mL) containing the EO at concentrations 10, 15, 20, and 25 mg/mL or AgNPs at 0.5, 1, 1.5, and 2 mg/L, and after that, they settled on two sheets of paper towels for soaking up the extra liquid. Then, each larvae group was put in petri dishes (9 × 1.5 cm) wrapped by two sheets of filter paper drenched with 1.0 mL of distilled water. In the groups receiving the combination of EPNs + EO/AgNPs, firstly the tick larvae were submerged in the EO/AgNPs solutions at the same previous concentrations for 1 min, after which they were moved to petri dishes and 1.0 mL of an aqueous solution having 500 IJs of EPNs was introduced. In the control groups, 200 larvae were immersed in an aqueous solution with 2% Tween 20 or distilled water for the same time and methods described above. For both the controls and the treated larvae, five repetitions were performed for each treatment/group (40 larvae/replicate). The petri dishes of all groups were sealed with parafilm to ensure the larvae did not escape and allow the EPN IJs to move around while keeping the filter paper lining wet. The dishes were then incubated in laboratory conditions at 25 ± 1 °C, RH of 75–80%. For calculation of the mortality rate, the number of dead larvae in each petri dish was counted for 4 days. By observation, it was established that larvae that moved their legs but were unable to walk were considered dead (Singh et al., 2018). Mortality percentage was corrected using Schneider-Orellis formula (Puntener, 1981) at each exposure period as follow:

CorrectedMortality(%)=(%MT−%MC100−%MC)×100

where MT: Mortality in Treatment, MC: Mortality in Control.

Adult ticks immersion test

On the engorged females, the bioassay experiment was carried out to evaluate the ixodicidal activity of EPNs applied singly and in combinations with EO or AgNPs. The adult immersion test of engorged H. dromedarii females (AIT) was conducted based on the procedure outlined by Monteiro et al. (2021). The experiment’s methodology was the same as for topic 2.6.2, with the exception that EPNs were applied at concentrations 500, 1,000, 1,500, and 2,000 IJs; EO of R. officinalis, at 20, 30, 40, and 60 mg/mL; while AgNPs concentrations were 1, 2, 3, and 4 mg/L. In the groups treated with EO or AgNPs, the females, that were weighed before the test were immersed in 10 ml of each tested concentration for 5 min. In combination groups, the engorged females were dipped in 10 mL of each concentration of EO or AgNPs for 5 min and subsequently put into petri dishes including EPNs (200 IJs/engorged female). Five repetitions of each concentration were performed (five engorged females/replicate). The dishes were then maintained in an incubator, which was adjusted to the same previous temperature and humidity, and daily mortality rate was recorded for 4 days.

Biological parameters of tick engorged females

Following the exposure period for each of the aforementioned groups, the engorged females that were still alive were taken out of the dishes, cleaned with distilled water, dried on paper towel sheets to remove any remaining water, weighed, and their initial weight (IW) recorded. They were then put into equal-sized dishes that had two filter paper sheets on them that had been saturated with 1.0 mL of distilled water for oviposition. The egg masses in each treatment were collected daily for 15 days. The egg mass deposited by each female was weighed individually (EW), transferred to labeled 10-ml modified syringes, with hydrophilic cotton sealing the distal portion cut off and placed again in the controlled growth chamber (25 ± 1 °C, 75–80% RH). The females were weighed once more following the oviposition period, and the residual weight (RW) was determined. Other biological parameters estimated were preoviposition period (POP), oviposition period (OP), survival period of females (SP), nutritional efficacy index (NEI), hatchability percentage (H%), egg productive index (EPI), inhibition of oviposition (IO%), reproduction efficiency index (REI), and the the control percentage (CP%) according to the formulas described by Alimi et al. (2021), Saman et al. (2023):

NEI=EW(mg)IW(mg)−RW(mg)×100

H(%)=NumberofhatchedlarvaeNumberoflaideggs×100

EPI=EW(mg)IW(mg)

IO(%)=EPIcontrolgroup−EPItreatedgroupEPIcontrolgroup×100

REI=EW(mg)×H%IW(mg)×20000∗

*Constant showing the amount of eggs in 1 g of laying eggs.

CP(%)=REIcontrol−REItreatedREIcontrol×100

Insecticidal activity test

The final (7th) instar of G. mellonella larvae weighing 180–200 mg were employed in the experiment. We utilized only highly mobile larvae without any black patches on their cuticle. The methodology of this experiment was the same for topic 2.6.2 in separated and combined groups, except that EPNs, were used at concentrations 250, 500, 1,000, and 2,000 IJs; EO of R. officinalis, at concentrations 20, 30, 40, and 60 mg/mL; whereas, AgNPs was utilized at 1, 2, 3, and 4 mg/L. In combination groups, EPNs were used at 100 IJs/larvae. Each concentration in all treated groups as well as controls contained 50 larvae (10 larvae/replicate) and all treatments were also repeated five times. Following treatment, larvae were incubated at the same previous mentioned conditions and their mortality was monitored daily up to 96 h.

Statistical analysis

The data were expressed using mean ± standard error (M ± SE). The mortality rates of EPNs, tick larvae and adults, and wax moth larvae were investigated using a two-way analysis of variance (ANOVA) and Duncan’s multiple range test. All analyses were carried out using the COSTAT software (Version 6.400). The concentrations causing 50% and 90% mortalities (LC50 and LC90) after 96 h exposure and time needs to kill 50% (LT50) of the tick and insect individuals for the EPNs, EO, and AgNPs applied singly or concomitantly were computed using Probit analysis (Finney, 1971) by applying the SPSS program (Version 23). The heterogeneity significance (P), the slope and intercept, and the χ2 values of the tested groups were adjusted at 0.05, and a 95% fiduciary confidence level was set up. Due to the non-normal distribution of the data, the nonparametric biological data were analyzed using the Kruskal–Wallis test and the Student–Newman–Keuls test to compare the means at a level of significance (P < 0.05).

Results

Chemical composition of R. officinalis EO

The GC-MS analysis presents the chemical constituents of rosemary EO, along with their retention times and peak area percentages, in Table 1. The results demonstrate the presence of twenty eight compounds, representing 82.4% of the total oil content. The major constituents were verbenone (bicyclo[3.1.1]hept-3-en-2-one, 4,6,6-trimethyl-, (1S)-) (23.80%), 2-(2′,4′-dimethoxyphenyl)-6-methoxy-benzofuran (10.26%), 2(1H)-phenanthrenone, 4a,9,10,10a-tetrahydro-6- hydroxy-1,1,4a-trimethyl-7-(1-methylethyl)-, (4aStrans)-(9.96%), demethylsalvicanol (5.12%), delta.9-tetrahydrocannabivarin (4.29%), ferruginol (3.22%), and geranyl formate (2.95%) with retention times of 6.08, 45.11, 41.29, 45.71, 40.71, 39.19, and 6.99 min, respectively, followed by 2,6-octadiene-1,8-diol, 2,6-dimethyl-with 2.40% and 12.76 min, and 13-isopropylpodocarpen-12-ol-20-al with 2.26% and 44.18 min. However, caryophyllene oxide, 2,6-dimethylocta-3,5,7-trien-2-ol, and methyleugenol recorded 2.05, 2.03, and 1.79%, respectively. Lupeol was the least abundant of all the compounds identified, with a 0.16% area coverage and a retention time of 61.04 min.

Table 1 GC-MS analysis of R. officinalis EO.

Peak	R.T.*	Compounds	Area (%)	
1	6.08	Bicyclo[3.1.1]hept-3-en-2-one, 4,6,6-trimethyl-, (1S)-	23.80	
2	6.32	2-Heptene, 5-ethyl-2,4-dimethyl-	0.96	
3	6.99	Geranyl formate	2.95	
4	7.86	1,7-Octadiene-3,6-diol, 2,6-dimethyl-	1.66	
5	8.35	Thymol	0.40	
6	9.98	Eugenol	0.32	
7	10.90	2,6-Dimethylocta-3,5,7-trien-2-ol	2.03	
8	11.61	Methyleugenol	1.79	
9	12.76	2,6-Octadiene-1,8-diol, 2,6-dimethyl-	2.40	
10	13.48	Acetophenone, 4′-hydroxy-	0.84	
11	14.36	Hydroxychavicol	0.96	
12	14.45	(1S,2S,4S)-Trihydroxy-p-menthane	0.65	
13	17.46	Caryophyllene oxide	2.05	
14	19.37	11,11-Dimethyl-4,8- dimethylenebicyclo[7.2.0]undecan-3-ol	1.40	
15	19.73	2-Butanone, 4-(4-hydroxy-3-methoxyphenyl)-	1.03	
16	30.85	Cannabidivarol	0.21	
17	37.42	4-Hydroxy-4′-methyldiphenylamine, O-trimethylsilyl ether	0.88	
18	39.19	Ferruginol	3.22	
19	40.23	12-O-Methylcarnosol	0.99	
20	40.71	delta.9-Tetrahydrocannabivarin	4.29	
21	41.29	2(1H)-Phenanthrenone, 4a,9,10,10a-tetrahydro-6- hydroxy-1,1,4a-trimethyl-7-(1-methylethyl)-, (4aStrans)-	9.96	
22	44.18	13-Isopropylpodocarpen-12-ol-20-al	2.26	
23	45.11	2-(2′,4′-Dimethoxyphenyl)-6-methoxy-benzofuran	10.26	
24	45.71	Demethylsalvicanol	5.12	
25	48.77	Androst-5-en-7-one, 3-(acetyloxy)- (3.beta.)-	0.54	
26	51.29	.alpha.-Tocospiro B	0.56	
27	56.11	dl-.alpha.-Tocopherol	0.60	
28	61.04	Lupeol	0.16	
Note:

* R.T., Retention time (min).

Characterization of P. mirabilis bacterium-synthesized AgNPs

This work described the biosynthesis of AgNPs by the P. mirabilis bacterium. Upon incubating the bacterial filtrate with silver nitrate at room temperature, we observed a visual color change from bright yellow to dark brown, while AgNO3alone did not exhibit any color change (Fig. 1A). AgNPs were found to be spherical and monodispersed, with an average diameter ranging from 5 to 45 nm, according to the TEM image and size distribution (Figs. 1B, 1C). Figure 1D shows the UV-vis spectrum of AgNP biosynthesis. Consequently, we discovered the largest surface plasmon peak of the colloidal solution’s AgNPs at 320 nm.

Figure 1 Characterization of silver nanoparticles.

(A) Color photo of silver nitrate (1), P. mirabilis supernatant (2), and AgNPs (3); (B) TEM image of AgNPs; (C) AgNPs’ particle size distribution; and (D) AgNPs’ UV-Vis Absorption Spectrum.

Nematicidal activity of EO or AgNPs against EPNs

Figure 2 summarizes the mortality rates of Steinernema sp. and H. indica dauer juveniles exposed to R. officinalis EO and biosynthesized AgNPs treatments at different concentrations and exposure periods. The results revealed significant differences (P < 0.05) between all concentrations examined against both EPN species at various exposure durations. All concentrations of EO or AgNPs were considered harmless to the two species of EPNs, as the treatments demonstrated less than 10% mortality. Long exposure time and high concentration of the EO (60 mg/mL) caused low mortality of both EPN species, which valued 5% for Steinernema sp. and 3.2% for H. indica (Figs. 2A, 2B). At the highest concentration of AgNPs (4 mg/L), however, Steinernema sp. died 9.8% of the time and H. indica juveniles died 4% of the time after 96 h of exposure (Figs. 2C, 2D).

Figure 2 Mortality percentage of EPNs.

Mortality percentage (±SE) of EPNs exposed to R. officinalis EO (A, B) and biosynthesized AgNPs (C, D) at four concentrations and exposure times. Significant variations between treatments for each period are indicated by different letters within each column (P < 0.05).

Efficiency of EPNs, EO, and AgNPs against H. dromedarii larvae

Table 2 shows the acaricidal activity of the treatments, either alone or in combination, on the death of H. dromedarii larvae. The treatment clearly worked better or worse depending on the concentration and length of exposure. For example, the percentage of dead H. dromedarii larvae that died after being exposed to EPNs, EO, or AgNPs alone at four concentrations or all at once for four exposure periods showed a range of 18.4% to 100%. Among all the tested treatments, results indicated that the highest corrected mortality percentage (94.7%) was significantly recorded by H. indica + EO treatment (P < 0.05), followed by Steinernema sp. + EO (87.1%), H. indica + AgNPs (85.8%), H. indica alone (80.2%), and Steinernema sp. + AgNPs (80.0%), which did not significantly differ, while Steinernema sp., EO, and AgNPs alone afforded lower mortalities, which were 65.3, 40.1, and 35.0%, respectively (Table 2). The tested EO at median and high concentrations (15–25 mg/mL) when mixed with H. indica was the most effective, killing all of the ticks at 96 h. After exposing H. dromedarii larvae to 25 mg/mL EO + H. indica for just 24 h, the percentage of mortality significantly reached 100.0%. Treatment with EPNs alone caused 99.0 and 89.0% mortality rates in the tick larvae when treated with 1,200 IJs of H. indica and Steinernema sp., respectively, for 96 h. The efficiency of the highest concentrations of both EO and AgNPs recorded remarkable mortalities 96 h-post treatment which valued 60.5 and 51.6%, respectively, whereas the lowest concentrations of them were not satisfactory; about <23% mortality was found after 24 h (Table 2).

Table 2 Acaricidal activity of EPNs, EO, and AgNPs on the camel tick, H. dromedarii larva in-vitro.

Treatments	Conc.	aMortality %	Treatment means	
24 h	48 h	72 h	96 h	
Steinernema sp. (IJs)	1,200	80.5 ± 2.3	82.6 ± 1.8	86.8 ± 0.0	89.0 ± 0.5	65.3e	
600	67.9 ± 1.7	69.5 ± 1.3	72.6 ± 0.6	75.8 ± 1.3	
300	57.9 ± 1.7	59.5 ± 0.6	62.6 ± 1.0	71.1 ± 0.8	
150	37.9 ± 2.0	39.5 ± 1.4	42.6 ± 1.3	48.4 ± 1.1	
H. indica (IJs)	1,200	89.5 ± 0.8	93.2 ± 0.6	95.8 ± 0.6	99.0 ± 0.6	80.2d	
600	81.1 ± 1.0	83.7 ± 1.0	88.4 ± 0.6	92.1 ± 0.0	
300	70.5 ± 1.0	72.1 ± 1.1	76.3 ± 1.2	82.1 ± 1.0	
150	59.0 ± 0.6	62.1 ± 0.6	65.3 ± 0.5	73.2 ± 0.5	
EO (mg/mL)	25	49.5 ± 1.5	52.1 ± 1.0	54.7 ± 0.5	60.5 ± 0.0	40.1f	
20	40.6 ± 1.7	43.7 ± 1.1	46.3 ± 0.6	50.5 ± 1.0	
15	30.0 ± 1.3	31.6 ± 0.8	34.7 ± 0.5	39.5 ± 0.8	
10	22.6 ± 1.1	25.3 ± 0.6	27.9 ± 0.5	31.6 ± 0.0	
AgNPs (mg/L)	2	43.2 ± 0.6	45.8 ± 1.1	48.4 ± 0.6	51.6 ± 1.1	35.0g	
1.5	33.2 ± 1.1	35.8 ± 0.6	38.4 ± 0.6	43.2 ± 0.6	
1	27.4 ± 0.6	29.5 ± 0.5	31.6 ± 0.0	35.8 ± 0.6	
0.5	18.4 ± 0.8	22.1 ± 0.6	25.3 ± 0.6	29.0 ± 0.8	
Steinernema sp. (500 IJs) + EO (mg/mL)	25	93.7 ± 0.6	94.7 ± 0.0	97.9 ± 1.0	100.0 ± 0.0	87.1b	
20	86.3 ± 1.0	87.9 ± 0.6	91.1 ± 0.6	95.8 ± 0.6	
15	80.0 ± 0.6	82.6 ± 0.6	85.3 ± 0.6	88.9 ± 0.5	
10	72.1 ± 1.1	73.7 ± 0.0	79.5 ± 0.5	84.2 ± 0.8	
H. indica (500 IJs) + EO
(mg/mL)	25	100.0 ± 0.0	100.0 ± 0.0	100.0 ± 0.0	100.0 ± 0.0	94.7a	
20	97.9 ± 1.3	100.0 ± 0.0	100.0 ± 0.0	100.0 ± 0.0	
15	87.9 ± 1.1	92.1 ± 0.8	96.3 ± 1.1	100.0 ± 0.0	
10	80.0 ± 0.6	82.1 ± 0.5	86.3 ± 0.5	93.2 ± 0.6	
Steinernema sp. (500 IJs) + AgNPs (mg/L)	2	90.0 ± 0.5	91.1 ± 0.6	93.7 ± 0.6	93.7 ± 0.6	80.0d	
1.5	82.6 ± 0.6	84.2 ± 0.0	87.4 ± 0.5	87.4 ± 0.5	
1	72.1 ± 1.1	74.7 ± 1.3	78.4 ± 2.1	78.4 ± 2.1	
0.5	63.7 ± 1.5	65.3 ± 1.0	68.4 ± 0.8	68.4 ± 0.8	
H. indica (500 IJs) + AgNPs
(mg/L)	2	92.1 ± 0.0	94.2 ± 0.5	97.4 ± 0.0	100.0 ± 0.0	85.8c	
1.5	83.7 ± 0.5	85.3 ± 0.6	89.0 ± 0.5	93.7 ± 0.6	
1	78.4 ± 0.5	80.5 ± 0.6	83.7 ± 1.0	87.9 ± 0.6	
0.5	72.1 ± 0.6	73.7 ± 0.0	77.9 ± 0.6	82.6 ± 0.6	
Exposure time means	67.0d	69.1c	72.2b	75.8a		
Notes:

a Five replicates, each containing 200 larvae, were used in this experiment to represent each treatment.

b The mortality ± standard error was denoted by the numbers in each column. Within a column or row, means with different letters show significant variations (P < 0.05 using Duncan’s Multiple Range Test).

The concentrations that kill 50% and 90% of H. dromedarii larvae after 96 h of exposure are shown in Table 3. H. indica alone was more effective than Steinernema sp. because its LC50 and LC90 values were much lower: 68.6 and 438.4 IJs for H. indica and 157.4 and 1,367.0 IJs for Steinernema sp. For EO alone and its mixture with EPNs, LC50 and LC90 values ranged from lowest to highest as follows: 3.9 and 8.7, 5.1 and 13.3, 19.0 and 44.8 mg/mL for H. indica + EO, Steinernema sp. + EO, and EO alone, respectively. AgNPs both individually and in combination, yielded LC50 and LC90 values of 0.1 and 1.1, 0.3 and 1.7, 1.9 and 4.1 mg/L for H. indica + AgNPs, Steinernema sp. + AgNPs, and AgNPs alone, respectively. The H. dromedarii larvae showed differing levels of homogeneity as a result of the studied treatments, with slope values varying from 0.97 to 3.65 (Table 3).

Table 3 The calculated LC50 and LC90 values of EPNs, EO, and AgNPs against H. dromedarii larvae.

Treatments	LC50
(95% LCL–UCL)	LC90
(95% LCL–UCL)	Slope ± SE	Intercept	X 2	P-value	
Steinernema sp. (IJs)	157.4 [89.1–199.6]	1367.0 [937.1–2,663.9]	1.33 ± 0.21	−2.87	6.18	0.046	
H. indica (IJs)	68.6 [29.4–105.8]	438.4 [343.2–627.0]	1.59 ± 0.29	−2.92	1.91	0.380	
EO (mg/mL)	19.0 [16.3–23.6]	44.8 [36.2–66.6]	1.89 ± 0.43	−2.41	0.85	0.653	
AgNPs (mg/L)	1.9 [1.6–2.8]	4.1 [3.7–10.1]	0.97 ± 0.29	−0.30	0.65	0.724	
Steinernema sp. (500 IJs) + EO (mg/mL)	5.1 [2.2–7.1]	13.3 [11.2–15.5]	3.08 ± 0.70	−2.19	4.02	0.134	
H. indica (500 IJs) + EO (mg/mL)	3.9 [0.2–6.3]	8.7 [3.6–10.7]	3.65 ± 1.32	−2.14	2.24	0.327	
Steinernema sp. (500 IJs) + AgNPs (mg/L)	0.3 [0.1–0.4]	1.7 [1.4–2.8]	1.62 ± 0.33	0.90	1.84	0.399	
H. indica (500 IJs) + AgNPs (mg/L)	0.1 [0.04–0.29]	1.1 [0.7–1.9]	1.77 ± 0.42	1.39	5.66	0.059	
Note:

LC50, lethal concentration capable of killing 50% of tick larvae; LC90, lethal concentration that kills 90% of tick larvae; LCL, lower confidence limit; UCL, upper confidence limit; X2, Chi-square value; SE, standard error and P-value, probability.

Efficacy of EPNs, EO, and AgNPs on mortality and biological measurements of H. dromedarii engorged females

For engorged females of H. dromedarii, the results shown in Table 4 illustrate that the average corrected mortality percentages varied from 0.0 to 100.0% when exposed to individual or combined treatments for four exposure times. Likewise, a significant increase in tick adult mortality (92.7%) was observed when EO was applied in association with H. indica compared to all other treatments. H. indica + AgNPs and Steinernema sp. + EO treatments, on the other hand, killed 81.3% and 80.7% of engorged H. dromedarii females, respectively; the differences between the percentages were not significant. Additionally, applying H. indica EPN alone (70.1%) and combining it with AgNPs (72.7%) resulted in statistically equal adult mortality. Using AgNPs alone resulted in the least significant mortality value of 26.4%. A statistical analysis of the adult mortality data shows significant differences (P < 0.05) between the mortality and the treatments at all concentrations from 24 to 96 h. When 60 mg/mL of EO was mixed with 1,000 IJs of each EPN species, all but one of the females that were fully grown died after 96 h of exposure (Table 4). Similarly, we observed a 100.0% mortality rate of the females using a mixture of H. indica (1,000 IJs) and EO (60 mg/mL) at 24, 48, and 72 h post applications. Among all single treatments tested, EPN H. indica at high concentrations (2,000 IJs) was the most toxic against engorged females, with a 95.8% mortality percentage, while higher concentrations of AgNPs (4 mg/L) showed less toxicity, with 50.0% mortality at 96 h exposure (Table 4). However, a low concentration of EO (20 mg/mL) treatment caused only 16.7% of mortality at 24 h after application, while the lowest concentration of AgNPs (1 mg/L) showed no mortality after 24 and 48 h of treatment.

Table 4 Acaricidal activity of EPNs, EO, and AgNPs on H. dromedarii engorged females under laboratory condition.

Treatments	Conc.	aMortality %	Treatment means	
24 h	48 h	72 h	96 h	
Steinernema sp. (IJs)	2,000	62.5 ± 4.2	66.7 ± 5.1	75.0 ± 4.2	79.2 ± 0.0	55.5d	
1,500	50.0 ± 5.1	58.3 ± 0.0	66.7 ± 5.1	70.8 ± 5.1	
1,000	41.7 ± 4.2	50.0 ± 5.1	58.3 ± 6.6	62.5 ± 4.2	
500	25.0 ± 5.1	33.3 ± 4.2	41.7 ± 4.2	45.8 ± 5.1	
H. indica (IJs)	2,000	75.0 ± 4.2	83.3 ± 4.2	91.7 ± 5.1	95.8 ± 4.2	70.1c	
1,500	62.5 ± 4.2	70.8 ± 5.1	79.2 ± 0.0	83.3 ± 4.2	
1,000	54.2 ± 4.2	58.3 ± 0.0	66.7 ± 5.1	70.8 ± 5.1	
500	45.8 ± 5.1	54.2 ± 4.2	62.5 ± 4.2	66.7 ± 5.1	
EO (mg/mL)	60	45.8 ± 5.1	54.2 ± 4.2	62.5 ± 7.8	66.7 ± 5.1	42.7e	
40	37.5 ± 0.0	45.8 ± 5.1	54.2 ± 4.2	58.3 ± 0.0	
30	25.0 ± 5.1	33.3 ± 4.2	41.7 ± 4.2	45.8 ± 5.11	
20	16.7 ± 0.0	25.0 ± 5.1	33.3 ± 4.2	37.5 ± 0.0	
AgNPs (mg/L)	4	33.3 ± 4.2	41.7 ± 7.8	45.8 ± 8.3	50.0 ± 8.3	26.4f	
3	25.0 ± 5.1	33.3 ± 4.2	33.3 ± 4.2	37.5 ± 0.0	
2	13.3 ± 3.3	21.7 ± 7.1	25.0 ± 5.1	29.2 ± 5.1	
1	0.0 ± 0.0	0.0 ± 0.0	16.7 ± 0.0	16.7 ± 0.0	
Steinernema sp. (1,000 IJs) + EO (mg/mL)	60	87.5 ± 5.1	91.7 ± 5.1	95.8 ± 4.2	100.0 ± 0.0	80.7b	
40	79.2 ± 0.0	83.3 ± 4.2	83.3 ± 4.2	87.5 ± 5.1	
30	70.8 ± 5.1	75.0 ± 4.2	79.2 ± 0.0	83.3 ± 4.2	
20	62.5 ± 4.2	66.7 ± 5.1	70.8 ± 5.1	75.0 ± 4.2	
H. indica (1,000 IJs) + EO
(mg/mL)	60	100.0 ± 0.0	100.0 ± 0.0	100.0 ± 0.0	100.0 ± 0.0	92.7a	
40	91.7 ± 5.1	100.0 ± 0.0	100.0 ± 0.0	100.0 ± 0.0	
30	79.2 ± 0.0	87.5 ± 5.1	95.8 ± 4.2	100.0 ± 0.0	
20	70.8 ± 5.1	79.2 ± 0.0	87.5 ± 5.1	91.7 ± 5.1	
Steinernema sp. (1,000 IJs) + AgNPs
(mg/L)	4	83.3 ± 4.2	87.5 ± 5.1	87.5 ± 5.1	91.7 ± 5.1	72.7c	
3	70.8 ± 5.1	79.2 ± 0.0	83.3 ± 4.2	87.5 ± 5.1	
2	62.5 ± 4.2	66.7 ± 5.1	66.7 ± 5.1	70.8 ± 5.1	
1	50.0 ± 5.1	54.2 ± 4.2	58.3 ± 0.0	62.5 ± 4.2	
H. indica (1,000 IJs) + AgNPs
(mg/L)	4	91.7 ± 5.1	95.8 ± 4.2	100.0 ± 0.0	100.0 ± 0.0	81.3b	
3	79.2 ± 0.0	87.5 ± 5.1	87.5 ± 5.1	91.7 ± 5.1	
2	70.8 ± 5.1	75.0 ± 4.2	75.0 ± 4.2	79.2 ± 6.6	
1	62.5 ± 4.2	66.7 ± 5.1	66.7 ± 5.1	70.8 ± 5.1	
Exposure time means	57.1d	63.3c	68.5b	72.1a		
Notes:

a Each treatment in this experiment was represented by five replicates, each with 25 engorged females.

b Numbers in each column indicated to mortality ± standard error.

Means with different letters within the column or row differ significantly (P < 0.05 using Duncan’s Multiple Range Test).

As shown in Table 5, probit analysis exposed the LC50, LC90, confidence interval limits at 95%, slope value, intercept, Chi-square, and probability. Similarly, the data indicate that H. indica was significantly more virulent against H. dromedarii engorged females than Steinernema sp. with LC50 (391.8 and 577.9 IJs) and LC90 (1,760.6 and 2,736.2 IJs), respectively. The treatment of EO combined with H. indica (1,000 IJs) exhibited lower LC50 value of 9.6 mg/mL than Steinernema sp. (1,000 IJs) + EO (12.8 mg/mL), and EO alone (34.3 mg/mL), whereas, it recorded 0.6, 0.7, and 4.2 mg/L for H. indica + AgNPs, Steinernema sp. + AgNPs, and AgNPs alone, respectively. The highest degree of homogeneity in individuals of H. dromedarii engorged females’ response was noticed after exposure to the H. indica + EO treatment, with a slope value of 3.14.

Table 5 The calculated LC50 and LC90 values of EPNs, EO, and AgNPs against H. dromedarii engorged females.

Treatments	LC50
(95% LCL–UCL)	LC90
(95% LCL–UCL)	Slope ± SE	Intercept	X 2	P-value	
Steinernema sp. (IJs)	577.9 [90.8–837.2]	2,736.2 [2,264.0–3,772.5]	1.46 ± 0.29	−4.06	0.67	0.718	
H. indica (IJs)	391.8 [224.4–532.4]	1,760.6 [1,517.2–2,204.7]	1.64 ± 0.32	−4.11	9.34	0.009	
EO (mg/mL)	34.3 [26.1–41.2]	102.1 [81.2–156.2]	1.62 ± 0.37	−2.43	0.78	0.677	
AgNPs (mg/L)	4.2 [3.4–6.9]	8.1 [6.6–11.6]	1.56 ± 0.31	−0.99	0.50	0.778	
Steinernema sp. (1,000 IJs) + EO (mg/mL)	12.8 [5.7–18.2]	44.5 [37.6–67.9]	2.79 ± 0.54	−3.06	6.28	0.043	
H. indica (1,000 IJs) + EO (mg/mL)	9.6 [4.3–11.9]	17.5 [6.9–22.0]	3.14 ± 1.13	−2.62	3.31	0.191	
Steinernema sp. (1,000 IJs) + AgNPs (mg/L)	0.7 [0.4–1.0]	3.9 [3.0–6.1]	1.76 ± 0.32	0.25	3.98	0.137	
H. indica (1,000 IJs) + AgNPs (mg/L)	0.6 [0.2–0.9]	2.7 [2.2–3.0]	2.22 ± 0.37	0.43	9.73	0.008	
Note:

LC50, lethal concentration that kills 50% of tick females; LC90, lethal concentration that kills 90% of tick females; LCL, lower confidence limit; UCL, upper confidence limit; X2, Chi-square value; SE, standard error and P-value, probability.

Exposure to treatments impacted all biological characteristics in engorged females of H. dromedarii (Table 6). Significant differences (P < 0.05) were observed between the treated groups and controls in all of the biological parameters measured, even though there were no significant differences in some biological parameters evaluated among the treated groups. The average initial weight of engorged females ranged from 61.3 to 543.2 mg, and no differences (P < 0.05) were noticed between treated groups when exposed to dual applications and single treatment of H. indica. Results for the treatment and control groups differed significantly with regard to the pre- and oviposition periods of engorged H. dromedarii females (P < 0.05). In all groups, the pre-oviposition period varied from 1.0 to 4.4 days. The treated groups’ oviposition periods were significantly shorter (P < 0.05) and only took 2, and 2.4 days when H. indica was used with either EO or AgNPs, compared to 3.0 days when H. indica was used by itself. There were some small but significant differences between the treatments. In terms of egg mass weight, treatment exposure interfered with the oviposition of treated engorged H. dromedarii females. In the treated groups, this was significantly lower for engorged females than in the untreated ones (P < 0.05). The best treated group was H. indica + EO, which significantly decreased the egg mass weight (P < 0.05) by 16.2 mg. It was followed by H. indica + AgNPs, which decreased it by 28.4 mg, and then Steinernema sp. + EO, which decreased it by 29.4 mg. The highest egg mass weight (175.4 mg) was recorded from AgNPs alone when compared to the control groups, which had weights of 421.1 mg of distilled water and 314.8 mg of tween 20 (Table 6). The egg hatching percentage likewise showed substantial variation (P < 0.05) between the treated and untreated groups; values in the treated groups varied from 11.5 to 53.1%, while the distilled water group had a value of 100.0%. The survival period varied significantly among different treatments, ranging from 4.0 to 9.8 days in the treated groups, while it recorded 19.0 and 18.3 days in the control groups. We did not observe any statistical differences between all combination treatments, but we did observe a significant difference in the nutritional efficiency index between all the treated groups and the controls. The distilled water group obtained a NEI value of 86.9%, significantly higher than all other treated groups, which ranged from 28.6% to 56.7%. The association of H. indica and EO resulted in the highest significant inhibition of oviposition (IO%) among all treatments, with 73.2%, while engorged females exposed to AgNPs alone showed the least inhibition, with 28.9%. The best control percentage was of 95.6% obtained in the group treated with H. indica + EO, followed by H. indica + AgNPS, Steinernema sp. + AgNPs, Steinernema sp. + EO, and H. indica alone with 92.6, 91.9, 90.6, and 89.4%, despite not being significantly different among them. Conversely, AgNPs alone demonstrated a lower level of control over H. dromedarii females, with a rate of 62.8%.

Table 6 Biological parameters of H. dromedarii engorged females exposed to EPNs, EO, and AgNPs either singly or in combinations under laboratory conditions.

Treatments
a(n)	bBiological parameters	
IW(mg)	POP(days)	OP(days)	EW(mg)	H(%)	SP(days)	NEI(%)	IO(%)	CP(%)	
Steinernema sp. (15)	207.1e ± 3.3	2.6d ± 0.1	4.3d ± 0.2	108.9e ± 2.4	28.9d ± 0.4	7.8e ± 0.2	53.1c ± 0.8	34.2d ± 2.4	77.3b ± 1.1	
H. indica (10)	102.1f ± 2.9	1.8ef ± 0.1	3.0ef ± 0.2	34.6f ± 0.8	20.7e ± 0.8	5.6f ± 0.2	37.1d ± 1.3	58.1b ± 1.6	89.4a ± 0.6	
EO (20)	258.0d ± 5.8	3.1c ± 0.1	5.0d ± 0.1	139.5d ± 2.9	39.5c ± 1.0	8.4d ± 0.1	55.2c ± 1.2	29.2e ± 1.6	70.7c ± 1.5	
AgNPs (25)	342.5c ± 4.9	3.7b ± 0.1	5.5c ± 0.1	175.4c ± 1.9	53.0b ± 0.9	9.8c ± 0.1	56.7c ± 1.5	28.9e ± 1.7	62.8d ± 1.2	
Steinernema sp. + EO (8)	89.3fg ± 2.3	1.4fg ± 0.2	2.8ef ± 0.2	29.4f ± 0.4	18.8ef ± 0.5	4.8fg ± 0.2	36.0d ± 1.1	50.1c ± 2.2	90.6a ± 0.6	
H. indica + EO (6)	61.3g ± 1.8	1.0g ± 0.0	2.0g ± 0.0	16.2g ± 0.3	11.5g ± 0.4	4.0g ± 0.0	28.6d ± 0.6	73.2a ± 0.4	95.6a ± 0.2	
Steinernema sp. + AgNPs (10)	105.2f ± 2.1	2.0e ± 0.0	3.2e ± 0.2	32.0f ± 0.5	19.3ef ± 0.4	5.4f ± 0.2	33.1d ± 0.9	58.1b ± 1.2	91.9a ± 0.5	
H. indica + AgNPs (8)	86.5fg ± 1.4	1.3g ± 0.2	2.4fg ± 0.2	28.4f ± 0.4	17.3f ± 0.4	4.1g ± 0.1	35.5d ± 0.6	58.2b ± 2.4	92.6a ± 0.3	
Tween 20 (28)	463.4b ± 8.7	4.4a ± 0.1	13.2b ± 0.1	314.8b ± 3.3	99.2a ± 0.2	18.3b ± 0.3	75.1b ± 1.8			
Distilled water (28)	543.2a ± 7.5	4.3a ± 0.1	14.4a ± 0.1	421.1a ± 4.8	100.0a ± 0	19.0a ± 0.2	86.9a ± 1.6			
Notes:

a Engorged females that survived and had their biological parameters measured were indicated by the numbers in the brackets.

b The mean ± standard error was given by the numbers in each column.

Kruskal-Wallis and Student-Newman-Keuls tests demonstrate that means with the same letters in each column do not substantially differ (P < 0.05). Initial weight (IW), preoviposition (POP), oviposition (OP), eggs weight (EW), larval hatching percentage (H), female survival period (SP), nutritional efficacy index (NEI), inhibition of oviposition (IO), and control percentage (CP).

Insecticidal activity of EPNs, EO, and AgNPs against G. mellonella larvae

In Table 7, it shows that the larvae of G. mellonella were highly sensitive (P < 0.05) to both types of EPNs, either by themselves or in combination with EO or AgNPs. The mean percentages of deaths were 97.1, 92.1, 92.0, 92.0, 87.9, and 82.6% for H. indica + EO, H. indica + AgNPs, H. indica alone, Steinernema sp. + EO, Steinernema sp. + AgNPs, and Steinernema sp. alone treatments, in that order. Because of their lower effectiveness, EO and AgNPs only caused 52.7% and 42.5%, respectively, of larval mortality (P < 0.05). We also observed a direct correlation between the treatment concentrations and the percentage mortality. Hence, the mortality rate increased in proportion to an increase in treatment concentration at different exposure times. In Table 7, we can see that high (60 mg/mL) and median (40 mg/mL) concentrations of EO were linked to 1,000 IJs of H. indica and high (100% mortality) rates for all counting days. However, the same treatment at low concentrations (20 and 30 mg/mL) caused 100.0% mortality 96 h-post application. Similarly, exposure of G. mellonella larvae for 48, 72, and 96 h to H. indica (2,000 IJs) alone, Steinernema sp. (1,000 IJs) + EO (60 mg/mL), and H. indica (1,000 IJs) + AgNPs (4 mg/L) resulted in 100.0% mortality. Single application of either EO or AgNPs at low concentrations, on the other hand, caused the lowest death rates (33.3 and 23.0%, respectively), compared to 75.0 and 62.5% from H. indica and Steinernema sp. after 24 h of exposure.

Table 7 Insecticidal activity of EPNs, EO, and AgNPs on the wax moth, G. mellonella larvae under laboratory condition.

Treatments	Conc.	aMortality %	Treatment means	
24 h	48 h	72 h	96 h	
Steinernema sp. (IJs)	2,000	81.3 ± 3.9	89.6 ± 0.0	93.8 ± 2.6	98.0 ± 2.1	82.6d	
1,000	79.2 ± 0.0	85.4 ± 2.6	89.6 ± 3.3	93.8 ± 4.2	
500	72.9 ± 2.6	81.3 ± 2.1	85.4 ± 2.6	89.6 ± 3.3	
250	62.5 ± 2.6	68.8 ± 0.0	73.0 ± 2.6	77.1 ± 2.1	
H. indica (IJs)	2,000	93.8 ± 2.6	100.0 ± 0.0	100.0 ± 0.0	100.0 ± 0.0	92.0b	
1,000	91.7 ± 2.1	95.8 ± 2.6	100.0 ± 0.0	100.0 ± 0.0	
500	83.3 ± 2.6	89.6 ± 0.0	93.8 ± 2.6	98.0 ± 2.1	
250	75.0 ± 2.6	79.2 ± 0.0	83.3 ± 2.6	87.5 ± 3.9	
EO (mg/mL)	60	60.4 ± 2.1	64.6 ± 2.6	68.8 ± 4.7	75.0 ± 4.2	52.7e	
40	52.1 ± 2.6	56.3 ± 2.1	60.4 ± 3.9	64.7 ± 2.6	
30	39.6 ± 2.1	43.8 ± 2.6	48.0 ± 0.0	52.1 ± 2.6	
20	33.3 ± 2.6	37.5 ± 0.0	41.7 ± 2.6	45.8 ± 2.1	
AgNPs (mg/L)	4	50.0 ± 2.1	45.2 ± 2.6	58.3 ± 0.0	62.5 ± 2.6	42.3f	
3	41.7 ± 2.6	43.8 ± 2.6	48.0 ± 3.3	52.1 ± 4.2	
2	31.3 ± 2.6	35.4 ± 2.1	39.6 ± 3.9	43.8 ± 2.6	
1	23.0 ± 2.6	27.1 ± 0.0	31.3 ± 2.6	35.4 ± 2.1	
Steinernema sp. (1,000 IJs) + EO (mg/mL)	60	95.8 ± 2.6	100.0 ± 0.0	100.0 ± 0.0	100.0 ± 0.0	92.0b	
40	89.6 ± 0.0	95.8 ± 2.6	100.0 ± 0.0	100.0 ± 0.0	
30	83.3 ± 2.6	89.6 ± 0.0	93.8 ± 2.6	98.0 ± 2.1	
20	75.0 ± 2.6	79.2 ± 0.0	83.3 ± 2.6	87.5 ± 2.1	
H. indica (1,000 IJs) + EO
(mg/mL)	60	100.0 ± 0.0	100.0 ± 0.0	100.0 ± 0.0	100.0 ± 0.0	97.1a	
40	100.0 ± 0.0	100.0 ± 0.0	100.0 ± 0.0	100.0 ± 0.0	
30	91.7 ± 2.1	95.8 ± 2.6	100.0 ± 0.0	100.0 ± 0.0	
20	83.3 ± 2.6	89.6 ± 3.3	93.8 ± 4.2	100.0 ± 0.0	
Steinernema sp. (1,000 IJs) + AgNPs
(mg/L)	4	93.8 ± 2.6	95.8 ± 2.6	100.0 ± 0.0	100.0 ± 0.0	87.9c	
3	85.4 ± 2.6	91.7 ± 2.1	93.8 ± 2.6	98.0 ± 2.1	
2	83.3 ± 2.6	85.4 ± 2.6	89.6 ± 0.0	93.8 ± 2.6	
1	70.8 ± 3.9	70.8 ± 3.9	75.0 ± 6.3	79.2 ± 4.7	
H. indica (1,000 IJs) + AgNPs
(mg/L)	4	95.8 ± 2.6	100.0 ± 0.0	100.0 ± 0.0	100.0 ± 0.0	92.1b	
3	89.6 ± 0.0	95.8 ± 2.6	100.0 ± 0.0	100.0 ± 0.0	
2	83.3 ± 2.6	87.5 ± 2.1	91.7 ± 2.1	95.8 ± 2.6	
1	77.1 ± 2.1	81.3 ± 2.1	85.4 ± 2.6	89.6 ± 0.0	
Exposure time means	74.0d	78.5c	81.9b	84.9a		
Notes:

a Each treatment in this experiment was represented by five replicates, each with 50 larvae.

b Numbers in each column indicated to mortality ± standard error. Means with different letters within the column or row differ significantly (P < 0.05 using Duncan’s Multiple Range Test).

Likewise, data in Table 8 confirmed that the IJs of H. indica were the most effective against G. mellonella larvae than Steinernema sp., with LC50 values of 107.2 and 173.1 IJs and LC90 values of 272.1 and 590.3 IJs, respectively, despite not being significantly different two EPNs. The calculated LC50 and LC90 values of EO + H. indica (1,000 IJs), Steinernema sp. (1,000 IJs) + EO, and EO alone were 6.8 and 12.7, 9.2 and 21.1, and 24.6 and 88.6 mg/mL, respectively; however, it exhibited 0.3 and 1.0, 0.6 and 1.7, and 2.7 and 7.1 mg/L for H. indica + AgNPs, Steinernema sp. + AgNPs, and AgNPs alone, respectively. The population of G. mellonella larvae exhibited the greatest degree of homogeneity for EO + H. indica and EO + Steinernema sp. treatments with slope values of 3.83 and 3.66, respectively, whereas, with the exception of H. indica, all of the individual treatments had low slope values, showing heterogeneity in the insect reaction to these treatments (Table 8).

Table 8 Median lethal numbers (LC50, LC90) of EPNs, EO, and AgNPs against G. mellonella larvae.

Treatments	LC50
(95% LCL–UCL)	LC90
(95% LCL–UCL)	Slope ± SE	Intercept	X 2	P-value	
Steinernema sp. (IJs)	173.1 [118.5–234.2]	590.3 [442.7–848.1]	1.42 ± 0.30	−2.64	2.51	0.285	
H. indica (IJs)	107.2 [43.9–162.2]	272.1 [198.5–336.3]	3.17 ± 0.95	−6.44	0.17	0.921	
EO (mg/mL)	24.6 [12.6–31.5]	88.6 [72.0–129.5]	1.69 ± 0.37	−2.34	0.82	0.663	
AgNPs (mg/L)	2.7 [1.8–3.3]	7.1 [6.3–13.3]	1.13 ± 0.28	−0.43	0.98	0.612	
Steinernema sp. (1,000 IJs) + EO (mg/mL)	9.2 [7.1–22.7]	21.1 [15.8–24.7]	3.66 ± 1.00	−3.56	1.80	0.406	
H. indica (1,000 IJs) + EO (mg/mL)	6.8 [4.4–15.1]	12.7 [7.2–18.3]	3.83 ± 0.65	−2.57	4.32	0.785	
Steinernema sp. (1,000 IJs) + AgNPs (mg/L)	0.6 [0.3–1.0]	1.7 [1.4–2.0]	2.85 ± 0.52	0.78	0.94	0.626	
H. indica (1,000 IJs) + AgNPs (mg/L)	0.3 [0.1–0.6]	1.0 [0.7–1.3]	2.63 ± 0.70	1.23	2.57	0.276	
Note:

LC50, lethal concentration that kills 50% of wax moth larvae; LC90, lethal concentration that kills 90% of wax moth larvae; LCL, lower confidence limit; UCL, upper confidence limit; X2, Chi-square value; SE, standard error and P-value, probability.

Lethal time of EPNs, EO, and AgNPs in the mortality of larvae and engorged females of H. dromedarii, and larvae of G. mellonella

The combined treatments required less time to kill 50% of H. dromedarii larvae, engorged females, and G. mellonella larvae than the individual ones (Table 9). It had a lower LT50 value of 10.6 h, which meant that G. mellonella larvae were more likely to be killed by H. indica than H. dromedarii larvae and females. However, H. indica + EO had an LT50 of 8.2, 9.7, and 10.8 h, while H. indica + AgNPs had the LT50 of 10.0, 11.5, and 13.5 h for wax moth larvae, tick larvae, and female ticks that are full of eggs, respectively. Table 9 also demonstrates that exposure to H. indica + EO resulted in the highest degree of homogeneity in the individual response, with slope values of 1.61, 1.0, and 1.50 for G. mellonella and H. dromedarii larvae and females, respectively. On the other hand, the low slope values of the other evaluated treatments demonstrated consistent responses from individual insects and ticks to these treatments.

Table 9 LT50 of EPNs, EO, and AgNPs against larvae and engorged females of H. dromedarii, and larvae of G. mellonella.

Treatments	H. dromedarii larvae	H. dromedarii engorged females	G. mellonella larvae	
LT50 (h)
(95% LCL–UCL)	Slope ± SE	X 2	P-value	LT50 (h)
(95% LCL–UCL)	Slope ± SE	X 2	P-value	LT50 (h)
(95% LCL–UCL)	Slope ± SE	X 2	P-value	
Steinernema sp.	19.1 [12.2–26.9]	0.42 ± 0.29	0.44	0.802	36.4 [15.2–50.9]	0.85 ± 0.28	0.22	0.898	14.3 [8.6–19.4]	0.97 ± 0.32	0.20	0.904	
H. indica	14.7 [8.2–19.0]	0.64 ± 0.32	0.70	0.706	20.5 [13.4–28.1]	0.96 ± 0.29	0.35	0.839	10.6 [5.5–16.9]	1.14 ± 0.40	0.17	0.918	
EO	111.2 [87.1–132.4]	0.39 ± 0.28	0.25	0.883	85.3 [61.7–203.0]	0.91 ± 0.29	0.11	0.947	46.7 [22.4–65.1]	0.52 ± 0.28	0.20	0.905	
AgNPs	124.5 [100.2–165.1]	0.40 ± 0.29	0.15	0.928	154.9 [112.7–420.8]	0.83 ± 0.31	0.04	0.981	105.7 [87.2–256.4]	0.50 ± 0.28	0.16	0.924	
Steinernema sp. + EO	12.6 [9.0–17.2]	0.69 ± 0.35	0.79	0.672	14.2 [9.0–18.8]	0.66 ± 0.32	0.29	0.867	10.7 [6.0–18.3]	1.14 ± 0.40	0.17	0.918	
H. indica + EO	9.7 [4.9–15.8]	1.00 ± 0.46	0.86	0.650	10.8 [6.6–17.1]	1.50 ± 0.42	0.43	0.806	8.2 [3.1–14.8]	1.61 ± 0.57	1.40	0.497	
Steinernema sp. + AgNPs	17.3 [11.5–24.6]	0.32 ± 0.18	0.11	0.954	18.3 [11.0–27.7]	0.53 ± 0.30	0.12	0.942	11.3 [9.7–15.2]	0.75 ± 0.36	0.50	0.779	
H. indica + AgNPs	11.5 [10.4–14.9]	0.66 ± 0.34	0.77	0.680	13.5 [10.0–19.2]	0.54 ± 0.32	0.17	0.919	10.0 [5.1–15.7]	1.09 ± 0.40	0.22	0.895	
Note:

LT50, lethal time that required to kill 50% of individuals; LCL, lower confidence limit; UCL, upper confidence limit; X2, Chi-square value; SE, standard error and P-value, probability.

Discussion

This study investigates, for the first time, the combined effects of endogenous entomopathogenic nematodes with either rosemary EO or P. mirabilis-AgNPs against H. dromedarii and G. mellonella. The results showed that applying EPNs alone was lethal to both pests; however, combining EPNs with EO or AgNPs sequentially improved the level of lethality in a synergistic or additive mode. All concentrations of EO and AgNPs were safe for the two species of EPNs that were tested (H. indica and Steinernema sp.), according to the conventional mortality bioassay test used in this study. Notably, Steinernema sp. exhibited significantly higher sensitivity to both compounds compared to H. indica, and AgNPs proved to be more toxic to both EPNs than EO. The findings of the (Barua et al., 2020) study align with these results, confirming the toxicity of rosemary EO to plant parasitic nematodes. While it doesn’t kill slug parasitic and entomopathogenic nematodes, clove, thyme, garlic, and cinnamon oil also harm beneficial nematodes. Our results suggest that rosemary EO, which has demonstrated effectiveness against harmful nematodes without harming beneficial nematodes, is suitable for use in pest management. This study highlights several significant concerns about the application of rosemary essential oil in IPM programs, which also utilize nematode biological control agents. Heterorhabditis and Steinernema species share similar basic biological and physiological traits with plant parasitic nematodes, and they even inhabit similar soil ecosystems (Kenney & Eleftherianos, 2016). In spite of this, Uludamar (2023) found that rosemary oil at 10,000 ppm killed more than 92% of root-knot nematode, Meloidogyne chitwoodi second stage juveniles 24 h after exposure, while our study only found low rates of EPN mortality. The outer layer of EPNs and plant parasitic nematodes has lipids that dissolve in oils. This means that EOs might be able to get through this layer and affect how the organisms work (Stadler & Buteler, 2009). However, unlike most plant parasitic nematodes, EPN IJs have a double cuticle, which would give them an additional defense against these EOs. In line with our study, Kucharska et al. (2011) found that high concentrations (5 ppm) of AgNPs killed 99.6% of H. bacteriophora and 96% of S. feltiae on the fifth day of the experiment. Lower concentrations (0.5 ppm) killed only 4% of H. bacteriophora but 85% of S. feltiae.

Obtained data indicated that the combined applications all showed higher mortalities in the host species tested than single applications at the different concentrations and exposure times tested. Previously, Capinera (2008) showed that the toxicological interaction of certain chemicals with EPNs can lead to an increase in the mixture’s final toxicity (synergism), an additive increase in toxicity, or a decrease in toxicity (antagonism). As a result, in the current study, the interaction of the tested EPNs with EO or AgNPs appeared to exhibit a synergistic or additive effect on tick and insect mortality 4 days after treatment. It was the combination of H. indica at half concentration and EO at high concentrations that killed all the larvae and engorged females of H. dromedarii 24 h after treatment (100% mortality rate). On the other hand, the same treatment of H. indica with either a moderate or high concentration of EO killed all the larvae of G. mellonella during the same exposure period. Increasing biological activity against the target organism and lowering participant concentration are the two basic goals of employing synergistic combinations. Furthermore, the higher chemical complexity of combinations reduces the probability of selection for resistance and the subsequent generation of resistant populations (Jyoti et al., 2019).

Interestingly, H. indica IJs alone demonstrated exceptional virulence against all examined host species, surpassing the virulence of Steinernema sp. and other single treatments. It verified more than 91% mortality in H. dromedarii ticks starting on the third day and reaching 100% in just 2 days of G. mellonella exposure. Moreover, the larvae and adults of the tick, as well as the larvae of the wax moth, showed extreme vulnerability to this EPN’s infection, even when exposed in small quantities 4 days after exposure. Because lepidopteran larvae are extremely susceptible to infection by many isolates, this was already predicted. Researchers use this species as a model insect to study EPNs (Dolinski, 2006). According to their behavioral ecology, cruiser EPNs actively seek out insects by recognizing chemical cues released by their hosts, such as uric acid, CO2, excrement, and intestinal fluid. A specific species of arthropod may attract infectious juveniles more strongly due to the types and quantities of chemical signals they release (Bisch et al., 2015). The greater number of hosts and consequent increase in chemical signal emission are probably the reasons for H. indica’s superior efficacy against H. dromedarii in groups containing larvae or G. mellonella. This might have enhanced the activity of IJs, enabling them to go through the soil more quickly, which would have facilitated and expedited the host’s search, localization, and penetration activities (Monteiro et al., 2020).

We also observed the same trend with Steinernema sp., which ranked second among all separate treatments in terms of virulence effects against tick and insect individuals. Therefore, in the present study, EPNs seemed to be the important factor in tick and insect mortalities when applied in combination with EO or AgNPs. In addition, it was clear from our results that rosemary EO showed higher lethality against all host species than AgNPs. Even though the ixodicidal and insecticidal effects of the AgNPs by themselves were very good in this study, adding EPNs might make them much more effective and even make them more resistant to insects and ticks. The current research revealed a reduction in the toxic effect of nanomaterials evaluated against various stages of H. dromedarii (Benelli et al., 2017). This could be a result of the particles’ composition or the ecological and biological characteristics of ticks, which can change an individual’s susceptibility to nanoparticles relative to others. The P. mirabilis supernatant solution changed color from pale yellow to dark brown upon the addition of AgNO3. The naked eye could observe this, indicating the bio-reduction of Ag ions and the formation of AgNPs in the reaction solution. These results were in accordance with (Ejidike & Clayton, 2022; Ogunsilea et al., 2024) who reported that the synthesis of AgNPs from Daucus carota and Chromolaena odorata leaf extract was confirmed by a gradual change in color from colorless to darkish brown precipitate. This led to the discovery that the AgNPs in the mixed particles had a big absorption peak centered at 320 nm when the UV spectroscopic study was done. The TEM study showed that the particles were evenly spread out and had a diameter of 5–45 nm. We used P. mirabilis to biosynthesize silver nanoparticles, resulting in particles ranging in size from 35 to 100 nm, in agreement with the previous report by Yasr & AL-Ramahy (2022). Vasyliev et al. (2020) also agreed with the result. They said that the particles’ surface plasmon vibrations are what cause the brown color to be absorbed, and that the shape, size, dielectric medium, and chemical environment around the dispersed nanoparticles in water strongly affect surface plasmon absorption.

In this work, we used GC-MS analysis of R. officinalis essential oil to identify twenty eight phytocompounds significant to science, industry, and biology. There are a lot of different chemicals that make it up, but some of the most important ones are verbenone, 2-(2′,4′-dimethoxyphenyl)-6-methoxy-benzofuran, 2(1H)-phenanthrenone, 4a,9,10,10a-tetrahydro-6-hydroxy-1,1,4a-trimethyl-7-(1-methylethyl)-, (4aStrans)-, demethylsalvicanol, delta.9-tetrahydrocannabivarin, ferruginol, geranyl formate, 2,6-octadiene-1,8-diol, 2,6-dimethyl-, 13-isopropylpodocarpen-12-ol-20-al, caryophyllene oxide, 2,6-dimethylocta-3,5,7-trien-2-ol, and methyleugenol. Other chemicals are present in trace or low amounts. The results were consistent with the findings of Theyyathel et al. (2023), who reported that chemical profiling identified twenty-six chemicals in the methanolic leaf extract of R. officinalis. These substances include ethers, long-chain hydrocarbons, sterols, fatty acid esters, fatty alcohols, terpenes, and vitamin metabolites. Also our results were in harmony with those previous reported by Saleh et al. (2022) in Saudi Arabia, who stated the presence of 18 constituents in rosemary essential oil, representing 99.93% of the total oil content. The major constituents detected in oil were bornyl acetate (26.59%), eucalyptol (17.38%), camphor (10.42%), borneol (9.78%), beta-caryophyllene (7.80%) and α-pinene (3.85%), having strong antimicrobial activity against Staphylococcus aureus. In similar study by Martinez-Velazquez et al. (2011), they recorded that rosemary EO was rich in α-pinene (31.07%), verbenone (15.26%), and 1,8-cineol (14.2%), whereas Mexican oregano essential oil included thymol (24.59%), carvacrol (24.54%), p-cymene (13.6%), and γ-terpinene (7.43%); however, garlic essential oil was rich in diallyl trisulfide (33.57%), diallyl disulfide (30.93%), and methyl allyl trisulfide (11.28%).

With this regard, several bioactivities of R. officinalis EO have been documented, including insecticidal (Waliwitiya, Kennedy & Lowenberger, 2009) and acaricidal activity against two-spotted spider mite, Tetranychus urticae (Miresmailli & Isman, 2006). On the contrary, Laborda et al. (2013) found that rosemary and sage essential oils caused acute contact toxicity against T. urticae, although they showed no insecticidal activity against Ceratitis capitata. Furthermore, rosemary essential oil produced 85% R. microplus larval tick mortality at the higher concentrations (10% and 20%), while Mexican oregano and garlic essential oils had very similar activity, producing high mortality (90–100%) in all tested concentrations (Martinez-Velazquez et al., 2011). In the present study, the acricidal and insecticidal activities of rosemary EO could be attributed to the presence of verbenone, benzofuran, phenanthrenone, and tetrahydrocannabivarin compounds, which have been previously confirmed by other studies (Martinez-Velazquez et al., 2011; Bracalini, Florenzano & Panzavolta, 2024; Park et al., 2019; Díaz et al., 2023). In the same context, Martinez-Velazquez et al. (2011) indicated that the main chemical constituents of rosemary EO, which are probably responsible for acaricidal activity, are α-pinene, verbenone, and 1,8-cineol, compounds. Similarly, GC-MS analyses of R. officinalis and Salvadora persica essential oils showed that the main monoterpenes in both oils were 1,8-cineol, α-pinene, and β-pinene, although in markedly different proportions, suggesting that essential oils exhibit a great potential as substitute methods for controlling Ixodes ricinus ticks (Elmhalli et al., 2019).

Research has recently focused on the pharmacological attributes of ferruginol, a diterpene phenol, which include antimalarial, anticancer, antibacterial, and cardioprotective activities (Lim et al., 2022; Salih et al., 2022). Similar to this study, Fraga et al. (2005) also found that compounds called demethylsalvicanol and ferruginol, which were taken from the hairy roots of Salvia broussonetii, were moderately effective at stopping the Leptinotarsa decemlineata insect from feeding. After oral administration, none of these compounds had antifeedant or negative effects on Spodoptera littoralis ingestion or weight gain. Other chemicals, like Geranyl formate and Caryophyllene oxide, were also found in the essential oil of R. officinalis and in the essential oil of Pelargonium spp. in other studies. These chemicals are known to be toxic to Ae. aegypti larvae and to keep them away (Nararak et al., 2023).

Among all tested host individuals in the present study, G. mellonella larvae demonstrated high susceptibility to all treatments, exhibiting higher mortalities and lower LT50 when exposed to both EPNs incorporated with EO and AgNPs. This suggests that the compatibility of EPNs with EO or AgNPs exhibited stronger insecticidal activity than acaricidal effect. Furthermore, the present study reported the larvicidal and adulticidal effects of compatible EPN treatments with EO and AgNPs on H. dromedarii larvae and engorged females, respectively. Additionally, the larvae responded to the treatments more quickly than the engorged females, who moved slowly during the 2 days following treatment. They laid a normal egg mass and number, slightly lower than the control, but there was a significant variation in hatchability. Furthermore, the engorged females died on the 3rd day, particularly when exposed to the lower concentration of AgNPs alone; however, the larvae that received the same concentration showed mortality 24 h after treatment. The larger chitinized cuticle area of the engorged females, compared to the larvae’s less chitinized body area, could explain this observation.

Indeed, according to the biological characteristics examined, all of the treatments suggested in this study proved to be compatible with the action of EPNs, either by themselves or in combination, on H. dromedarii engorged females. The parameters showed that the groups treated with H. indica had shorter pre-oviposition, oviposition, and survival periods than the groups treated with Steinernema sp. Also, the values for EW, H%, and NEI% were higher in the Steinernema sp. treatments than in the H. indica treatments. This shows that H. indica is stronger than Steinernema sp., whether they are used alone or together. Our findings reveal that applying EPN H. indica in combination with EO inhibits egg-laying in H. dromedarii females by more than 73%, resulting in only 11.3% of eggs hatching. The infectivity tests in this study confirmed that H. indica is very strong against H. dromedarii. They also showed, for the first time, that these nematodes can kill ticks before they lay their eggs, whether they are used alone or with EO. When engorged females of H. dromedarii were exposed to H. indica, it changed the egg mass weight, but not in a way that was statistically significant when AgNPs were added. In other words, the entomopathogenic nematode infections did not get in the way of the metabolic conversion process or the production of the nutrients needed for egg formation before the eggs were laid. As a result, the best control percentage (95.6%) was obtained from the H. indica + EO treatment; it is not statistically different from the treatment when H. indica alone is present, although the efficacy was lower (89.4%). The results were better than those reported by Saman et al. (2023). They said that using 150 IJ/engorged females of H. indica LPP1 and H. bacteriophora HP88 led to an almost 90% reduction in the number and viability of eggs laid by engorged females of R. microplus. Also, these results are similar to what was reported by Albogami (2024), who said that the two species of Heterorhabditis are among the most dangerous to engorged H. dromedarii females, even more so than EPN from the Steinernema genus.

The results of the current preliminary study indicated that applying EPNs alone or integrated with either EO or AgNPs promise to provide superior efficacy against ticks and insects, but further research targeting other economically important tick or insect species is needed to elucidate their mode of action, side effects, and formulation development to confirm the method’s success in (livestock and apiary).

Conclusion

Researchers all over the world are investigating and introducing novel, safe, and efficient bio-agents as alternatives for chemical pesticides. This trend is influenced by a number of factors, including resistance, toxicity, availability, low cost, low pollution of the environment, and low side effects. It is worth noting that arthropod pests of economic and medical significance can be effectively and economically controlled with EPNs, EO, and bio-synthesized AgNPs. In order to manage H. dromedarii and G. mellonella, our work is the first to document the interaction of EPNs with either EO or AgNPs. Fortunately, the results showed that associating EPNs with EO was more effective than using AgNPs for controlling the two organism species that were being studied. The use of rosemary EO at a sub-lethal concentration that is compatible with EPN appears to be promising for efficiently causing toxicity in tick and insect pests.

We propose that impregnating tick and insect cuticles with R. officinalis EO or P. mirabilis-AgNPs, followed by nematode exposure, shows compatibility for livestock use. This approach, where treated camels are sprayed with EO or AgNPs and then exposed to nematodes in pasture, could allow nematodes to infect ticks post-detachment. Alternatively, combining EPNs with EOs/AgNPs directly on infested camels may enhance tick control and slow resistance development. Although successful EPN applications are simpler pest-host models, similar methods may suppress wax moth populations outside beehives, like in storage by dipping or spraying infested combs. Further research is essential to fully explore EPNs, EOs, and AgNPs for pest control.

Supplemental Information

Supplemental Information 1 Silver nanoparticles characterization.

Supplemental Information 2 Biological Parameters of Tick Females.

Supplemental Information 3 Camel tick mortality.

Supplemental Information 4 AgNPs UV-Visible absorption spectrum.

Here we show the absorbance of AgNPs, cell-free supernatant and AgNO3 respectively.

Supplemental Information 5 Tick female mortality.

Supplemental Information 6 Dataset of non targeted compounds from rosemary GC-MS.

Supplemental Information 7 Wax moth mortality.

Supplemental Information 8 Tick larvae mortality.

Supplemental Information 9 SS mortality.

Additional Information and Declarations

Competing Interests

Author Contributions

Data Availability

The authors declare that they have no competing interests.

Bander Albogami conceived and designed the experiments, prepared figures and/or tables, and approved the final draft.

Hadeer Darwish conceived and designed the experiments, authored or reviewed drafts of the article, and approved the final draft.

Akram Alghamdi performed the experiments, prepared figures and/or tables, authored or reviewed drafts of the article, and approved the final draft.

Ahmed BahaaEldin Darwish performed the experiments, prepared figures and/or tables, and approved the final draft.

Wafa Mohammed Al-Otaibi performed the experiments, authored or reviewed drafts of the article, and approved the final draft.

Mohamed A. Osman analyzed the data, prepared figures and/or tables, authored or reviewed drafts of the article, and approved the final draft.

Zamzam M. Al Dhafar analyzed the data, prepared figures and/or tables, authored or reviewed drafts of the article, and approved the final draft.

Abeer Mousa Alkhaibari performed the experiments, analyzed the data, authored or reviewed drafts of the article, and approved the final draft.

Abadi M. Mashlawi performed the experiments, authored or reviewed drafts of the article, and approved the final draft.

Fadi Baakdah analyzed the data, authored or reviewed drafts of the article, and approved the final draft.

Ahmed Noureldeen conceived and designed the experiments, analyzed the data, prepared figures and/or tables, authored or reviewed drafts of the article, and approved the final draft.

The following information was supplied regarding data availability:

The raw data are available in the Supplemental Files.

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
