# Peer review of "Acaricidal and insecticidal activities of entomopathogenic nematodes combined with rosemary essential oil and bacterium-synthesized silver nanoparticles against camel tick, Hyalomma dromedarii and wax moth, Galleria mellonella"

_PeerJ, doi:10.7717/peerj.18782_

## Round 0.1 · original submission · Major Revisions

Dear Authors,

As per the recommendations of our expert reviewers, the manuscript in present form attracts certain issues to be addressed (included in reviewer's comments). Please do the needful and resubmit asap.

All the best.

·

Basic reporting

The Abstract is kind of too kind, can it be revised.

Many of the biological names or species are not italicized. This is common throughout the whole manuscript.
E.g. Line 181-186 and others.


Authors should improved the introduction and experimental section must be supported by the following references.

Green synthesis of silver nanoparticles mediated by Daucus carota L.: antiradical, antimicrobial potentials, in vitro cytotoxicity against brain glioblastoma cells. Green Chemistry Letters and Reviews, 2022, 15(2), 297-310. DOI: https://doi.org/10.1080/17518253.2022.2054290.

Biosynthesis and optimization of AgNPs yield from Chromolaena Odorata leaf extract using response surface methodology (RSM). Physical Chemistry Research, 2024, 12(1), 21-31. https://doi.org/10.22036/pcr.2023.366212.2226.

Experimental design

2.2 Insect
How was this insert identified as Galleria...

Some of the abbreviation should be first written in full before abbreviating to give clear understanding.

.

Validity of the findings

Conclusions should be revised.

Additional comments

The author wrote the manuscript concisely but might requires some revisions to improved the quality.

·

Basic reporting

In general the manuscript is well written and structured according to the journal style.
Following aspects of the work would require authors’ attention.

Scientific names throughout the manuscript should be italicized.
Introduction, lines 90-93 – The correct term is “infest” not “infect”.
Introduction, lines 120-130. It is suggested that this information be deleted since it is not directly related to the fact that the moth is a pest in beekeeping.
Introduction, lines 202-203. “There are few or no studies”. If there are few, cite them. If there are no studies, state it as such.
Results, line 461. “At the highest concentration of AgNPs (4 mg/L),”.
Results, line 479. “ticks” not “insects”.
Tables 3, 5 and 8. Remove the term “pathogenicity” since its use is not appropriate. Modify the title of the tables accordingly.
Lines 519-521. Use the terms “toxic” and “toxicity” instead of “virulent” and “virulence”.
Lines 583-586. 92.1 value is for H. indica + AgNPs not for H. indica alone. Correct it.
Line 635. Use another term instead of “pathogenic”.
Line 637. Use another term instead of “virulence”.

Experimental design

Table 9. How were the data in Table 9 obtained? What were the experimental conditions? What concentrations were used in the individual and combined treatments to obtain these results?

Validity of the findings

Results, line 428. 20 compounds were detected in the essential oil. Together they add up to around 27% composition (percentage of area). Explain the reason for the low percentage obtained in said composition.
Lines 541-542, “…even though there were no significant differences among the treated groups…”. As shown in Figure 6, there were statistically significant differences in the biological parameters evaluated among the treated groups. Therefore said statement is not true.
Discussion, lines728-739 - The authors should compare the essential oil chemical composition results with related works and highlight the differences among oils. Additionally, the acaricidal activity of rosemary essential oil has already been reported in other species of ticks. It is suggested to cite these works. Some examples are shown.
http://www.bioone.org/doi/full/10.1603/ME10140
http://dx.doi.org/10.1016/j.indcrop.2013.04.011
https://doi.org/10.1007/s10493-019-00373-5

At the end of the discussion section it would be desirable to include a consideration of how EPNs could be applied alone or in combination, to tick-infested camels or moth-infested apiaries.

Reviewer 3 ·

Basic reporting

Basic English used slight improvement for better clarity

Experimental design

Well defined
rigorous methodology used

Validity of the findings

Robust data generated and statically analyzed
well compared and supported with relevant studies
Nicely concluded

Annotated reviews are not available for download in order to protect the identity of reviewers who chose to remain anonymous.

Reviewer 4 ·

Basic reporting

Generally, though the manuscript has important message, but still its need some improvement in English proficiency throughout. Its introduction is poorly written as some part of introduction is just like literature review/just coding the findings of previous studies which needs to be deleted & only relevant information should be coded . It should be divided in different paragraphs as well for easily understandable for scientific/academic community. It has economic importance of ticks in detail, but it has coded very limited economic importance of greater wax moth. Further, mostly the names of tested pests, bacteria & plants are not Italic & uniform throughout the manuscript. Few tables are poorly labeled as well & some raw data is missing as well in supplementary files such as only one Fig is given in these files & some raw data of tables is missing as well. So it should be improved throughout.

Experimental design

The research is original & within the scope of the Journal. All instigations were performed with high technical & ethical standards, but most sections of methodology are supported without relevant references, few models & formulas provided in this section are also without references. Section headings are not uniform as well such as few section headings are written in Italic & few are not Italic. There should be a uniform pattern of this section headings/sub heading. It needs a proper scientific revision.

Validity of the findings

Validity of the results is somewhat complex as data is poorly interpreted. Discussion is also poorly written as instead of comparing the finding of these studies with previous studies, its few portions seems like introduction or detail description about tested specimens. Its conclusion also needs improvement as it’s not according to the standard findings of the studies.

Additional comments

It should be revised throughout in its current form & in my opinion, if the manuscript experimental studies are restricted only to camel tick, Hyalomma dromedarii with proper improvement & revision and by skipping Galleria mellonella, it could be a valuable addition to the academic/researcher community.

---

## Round 0.2 · Minor Revisions

Dear Authors,
Our reviewers have appreciated your efforts to improve the manuscript, but it still has few points to be resolved (see the annotated pdf). Please make corrections and resubmit asap. All the best

·

Basic reporting

improved

Experimental design

improved

Validity of the findings

improved

Additional comments

Manuscript corrected accordingly

·

Basic reporting

No comment.

Experimental design

No comment.

Validity of the findings

No comment.

Additional comments

The authors have satisfactorily addressed the observations made in the original article and have made the appropriate modifications.
My only observation to the corrected manuscript is that the authors should update in the abstract the number of compounds detected by GC-MS (Table 1).

Reviewer 3 ·

Basic reporting

English improved as suggested.

Experimental design

Well explained expermental design.

Validity of the findings

Robust data generated and statically analyzed
well compared and supported with relevant studies
Nicely concluded.

Additional comments

Recommended changes have been well incorporated and should be accepted as such.

Reviewer 4 ·

Basic reporting

In general, the manuscript is well written & revised. However, it needs some minor corrections such as line 43 ticks and insect pests management, line 45, 46 & 47 sentence starting from “Biological Control should be rephrased, line 61 mortality instead of deaths, line 63 tested individuals instead of individuals, line 99, avermectin instead of ivermectin. Line 142-152 should be deleted. Line 210, ticks instead of tick, line 224-239, please code a reference for this procedure. 2.4.2. Essential oil extraction; please code a reference for this procedure, 2.4.3. GC–MS analysis of EO; please code a reference for this procedure, 2.5.2. Characterization of AgNPs; please code a reference for this procedure, 2.6.3. Adult Ticks Immersion test, line 848-860 should be improved.

Experimental design

The research is original & within the scope of the Journal. All instigations were performed with high technical & ethical standards, but few sections of methodology are still not supported with relevant references which should be addressed.

Validity of the findings

The results are improved in revised version but, discussion still needs a little bit improvement. Its conclusion still needs improvement as well and it should be purely according to the standard findings of the studies.

Additional comments

The revised manuscript is improved and minor corrections should be addressed before proceeding further.

---

## Round 0.3 · accepted · Accept

Dear Authors,

I inform you with pleasure that all the comments from our reviewers favor the manuscript in present form to be published. This is an academic acceptance and need few tasks to be completed before its publication. So, I advise you to be available for few days to complete all the requirements to avoid any further delay in publication.

All the best for your future submissions and thank you for considering PeerJ.

·

Basic reporting

No comment.

Experimental design

No comment.

Validity of the findings

No comment.

Additional comments

Authors made the suggested change.

Reviewer 4 ·

Basic reporting

The manuscript is improved & well revised. Line number 63 should be corrected, such as ticks and insects mortalities instead of tick and insect mortalities. Line number 865 should be corrected, such as tick (write the name of the tested tick) and insect (write the name of the tested insect) cuticles.

Experimental design

Improved and well written.

Validity of the findings

Improved and well written.

Additional comments

The revised manuscript is improved & corrected accordingly.